# Human Tuberculosis in Migrant and Autocthonous Patients: A Ten-Year Single-Centre Experience

**DOI:** 10.3390/pathogens14080824

**Published:** 2025-08-20

**Authors:** Isabel García Soriano, Mónica Romero, Isabel Gascón, Verónica Solves, Reyes Pascual, Philip Erick Wikman-Jorgensen

**Affiliations:** 1Infectious Diseases Unit, Department of Internal Medicine, Hospital General Universitario de Elda–FISABIO, 03600 Elda, Spain; isabel.garcia.soriano13@gmail.com (I.G.S.);; 2Department of Clinical Medicine, Miguel Hernández University, 03550 San Juan de Alicante, Spain; 3Microbiology Unit, Clinical Analysis Department, Hospital General Universitario de Elda, 03600 Elda, Spain

**Keywords:** *Mycobacterium tuberculosis*, emigrants and immigrants, immigration, multidrug-resistant tuberculosis

## Abstract

In recent years, migratory movements have increased. This study aimed to compare tuberculosis cases in migrant and autochthonous patients. We conducted a retrospective analytical cohort study of patients diagnosed with tuberculosis in the Elda Health District (Alicante, Spain) between 2013 and 2023. Of the 98 patients analyzed, 28 (29.6%) were migrants, predominantly male (65%), with a mean age of 35.6 years. Pulmonary tuberculosis was present in 82% of patients in both groups, and nine cases of drug-resistant tuberculosis were identified. No significant differences were observed between groups in treatment cure rates, mortality, or hospitalization. Unfavourable outcomes—a composite endpoint comprising mortality, treatment failure, and loss to follow-up—were more frequent in males and in patients with elevated C-reactive protein (CRP) levels (*p* = 0.033) or a higher CRP/albumin ratio. Migrants accounted for a substantial proportion of total TB cases and tended to be younger, with fewer comorbidities and lower rates of substance use. They showed a non-significant trend toward higher loss to follow-up and drug resistance. Overall, unfavourable outcomes were associated with elevated CRP levels and the CRP/albumin ratio.

## 1. Introduction

Tuberculosis (TB) is a communicable disease caused primarily by *Mycobacterium tuberculosis* and less frequently by *Mycobacterium bovis*, *Mycobacterium caprae*, or other atypical mycobacteria. It can affect almost any organ, although the pulmonary form is most common, with transmission typically occurring via airborne spread from person to person. In 2023, TB was responsible for an estimated 1.25 million deaths, making it the leading cause of mortality from a single infectious agent worldwide [1].

Most TB cases occur in Southeast Asia (45%), Africa (24%), and the Western Pacific region (17%), with lower proportions in the Eastern Mediterranean (8.6%), the Americas (3.2%), and Europe (2.1%) [1]. This global distribution reflects the influence of social determinants of health, as TB is strongly associated with low socioeconomic status, malnutrition, limited access to healthcare, unemployment, and low educational attainment [1,2].

In the European Union (EU) and European Economic Area (EEA), TB incidence rates are among the lowest worldwide, with fewer than 60,000 cases reported annually. However, notification rates vary widely—by more than 20-fold between countries—with Poland, Romania, and the United Kingdom together accounting for approximately 40% of all reported cases in the region [3].

The European Centre for Disease Prevention and Control (ECDC) classifies Spain as a low-incidence country for TB, with fewer than 20 cases per 100,000 inhabitants reported annually [4].

In recent years, forced migration has increased. In 2024 alone, more than 63,000 migrants arrived in Spain as a result of natural disasters, armed conflicts, and political persecution [5]. From a TB perspective, the country of origin is a key determinant of the prevalence of latent and active infections and influences the introduction of more virulent strains and drug-resistant patterns into host countries [6].

In low-incidence countries, TB rates vary markedly between population subgroups, whereas in high-incidence countries the disease tends to affect the population more uniformly [2,3].

Migrants may develop TB through reactivation of latent tuberculosis infection (LTBI) acquired prior to migration [7], may arrive with active TB, or may contract the infection after arrival due to poor living and working conditions [3]. The risk of developing TB is highest during the first two to three years after immigration and, although it can remain elevated for a longer period, it typically does not normalize until around ten years after arrival [5,6,7]. Screening and early detection of TB among migrants and refugees can help reduce the number of TB cases in foreign-born populations [8,9].

The objective of this study was to compare the clinical and epidemiological characteristics of migrant and native-born patients with active TB, with the aim of characterizing disease profiles in each subgroup and informing more effective strategies for detection, diagnosis, and treatment.

## 2. Material and Methods

We conducted a retrospective analytical cohort study of patients diagnosed with imported and autochthonous tuberculosis in the Elda Health Department (Alicante, southeastern Spain). This district serves 189,629 inhabitants, of whom 6.67% are migrants. We reviewed medical records of all patients diagnosed with TB between 2013 and 2023.

Inclusion criteria were age ≥ 18 years and microbiologically confirmed *Mycobacterium tuberculosis* infection. Exclusion criteria were latent TB infection, infection caused by other mycobacterial species, and age < 18 years.

Variables collected included epidemiological data, time since migration (continuous), risk factors, drug susceptibility, comorbidities, and treatment outcomes. Migrants were defined as foreign-born individuals. An unfavourable outcome was defined as a composite endpoint comprising death, loss to follow-up, or treatment failure.

Data were extracted from electronic medical records using the Orion Clinic and Abucasis software systems at Hospital General Universitario de Elda. Case identification was based on microbiological data provided by the hospital’s Microbiology Unit.

Tuberculosis diagnosis was established by direct smear microscopy using Ziehl–Neelsen staining, culture on liquid media (BACTEC MGIT©, Becton, Dickinson and Company, Franklin Lakes, NJ, USA) and solid media (Löwenstein–Jensen), and molecular amplification assays (GeneXpert MTB/RIF©, Cepheid, Sunnyvale, CA, USA), which also detect rifampicin resistance.

Drug susceptibility testing for first-line anti-tuberculosis drugs was performed using the BACTEC MGIT 960 system, including SIRE drugs and pyrazinamide; resistance was confirmed with molecular assays. Second-line drug susceptibility testing was conducted selectively only when first-line resistance was identified.

Strains were classified according to World Health Organization definitions as multidrug-resistant (MDR; resistant to isoniazid and rifampicin), pre-extensively drug-resistant (pre-XDR; resistant to isoniazid, rifampicin, and any fluoroquinolone), or extensively drug-resistant (XDR; additionally resistant to bedaquiline and/or linezolid) [10,11].

A descriptive statistical analysis was performed. Qualitative variables were expressed as absolute numbers and percentages (n, %), and quantitative variables as means with standard deviations for normally distributed data or medians with interquartile ranges (IQR) otherwise. For key variables, 95% confidence intervals (CIs) were calculated. Missing data were minimal and not imputed; analyses were conducted on available cases only.

Normality of distribution for quantitative variables was assessed using the Kolmogorov–Smirnov test. For hypothesis testing, Student’s *t*-test was applied to normally distributed continuous variables, and the Mann–Whitney U test to non-normally distributed variables. Categorical variables were compared using the Chi-square test or Fisher’s exact test, as appropriate. Odds ratios (ORs) with 95% CIs were calculated.

To reduce confounding, binary logistic regression was performed, including variables that showed statistical significance in the bivariate analysis. ORs and 95% CIs were reported, with statistical significance set to *p* < 0.05. The optimal threshold for continuous variables included in the regression was determined using a method that identifies the cut-point minimizing the Euclidean distance between the ROC curve and the point of perfect classification (0, 1), thereby achieving the best balance between sensitivity and specificity.

This study was conducted in accordance with the principles of Good Clinical Practice and the Declaration of Helsinki [11] and complied with current regulations on personal data protection and applicable legislation, including Law 41/2002 on patient autonomy and Law 14/2007 on biomedical research [11,12]. Approval was obtained from the Institutional Review Board of Elda General University Hospital (protocol number 2023/43PI TBCIM) on 12 March 2024. Informed consent was waived due to the retrospective design and use of data derived from routine clinical practice. No interventions or additional procedures were performed on patients.

All data were anonymized to prevent identification and unauthorized access, ensuring the protection of patient privacy and confidentiality in accordance with data protection legislation. All information was handled in compliance with Spain’s Organic Law 3/2018 of 5 December on the Protection of Personal Data and Guarantee of Digital Rights [13].

## 3. Results

### 3.1. General Cohort Description

During the study period, a total of 104 patients were diagnosed with tuberculosis (TB) in the Elda Health Department. Six were excluded because of age < 18 and thus 98 were included in the analysis (Figure 1). Of these, 66 (67%) were male, with a mean age of 44 years. A total of 81 cases (82%) presented with exclusively pulmonary TB. Sixty patients (61.2%) had a positive sputum test, 74 (75.5%) a positive real-time polymerase chain reaction (PCR) test and 97 (98.9%) had a positive culture. One patient had negative sputum and culture and was diagnosed based on the real-time PCR test and clinical presentation.

Among comorbidities, 6 patients (6%) were HIV-positive, 10 (10%) had chronic obstructive pulmonary disease (COPD), 7 (7%) had diabetes mellitus, 14 (14%) had an oncological or autoimmune disease, and 9 (9%) had a history of neurological disorders. Regarding toxic habits, 56 patients (57%) were smokers, 35 (35%) consumed alcohol, and 7 (7%) were intravenous drug users.

### 3.2. Migrant vs. Autochthonous Comparison

A comparative analysis of clinical and epidemiological characteristics between migrant and native-born TB patients is summarized in Table 1. A total of 29 patients (29.6%) were migrants. Among them, 19 (65%) were male. The mean age of migrant patients was 35.6 years (range: 21.3–50), significantly lower than the mean age of native-born patients, which was 48.03 years (range: 30.37–65.7) (*p* < 0.001). Most migrant patients originated from Eastern Europe (n = 12; 41.4%) or South America (n = 12; 41.4%), and 16 individuals (55.2%) had been residing in Spain for less than 5 years.

Among the migrant group, comorbidities were less frequent: 1 case (3.4%) of HIV infection, 1 (3.4%) of COPD, 2 (6.9%) had diabetes or an oncological/autoimmune disease, and 3 (10.3%) had neurological history. These proportions were all lower than those observed in the native population.

Regarding toxic habits, 46 non-migrant patients (66%) and 10 migrant patients (34%) were smokers (*p* = 0.007); 32 (46%) of the native-born population and 3 (10.3%) of the migrant population consumed alcohol (*p* = 0.01). All 7 intravenous drug users (10%) belonged to the non-migrant group.

In the laboratory parameters, no significant differences were observed between the two groups in terms of albumin levels or C-reactive protein (CRP).

In terms of anatomical site of infection, 24 migrant patients (82%) had exclusively pulmonary TB, a proportion similar to that of non-migrants.

Overall, 87 patients (88%) were cured, and 6 (6%) died. Hospital admission was required in 46 cases (46%). Treatment was interrupted in 5 patients (5%) due to loss to follow-up, and 10 patients (10%) experienced treatment failure.

Drug resistance to first-line anti-TB treatment was detected in 9 cases. Six cases were recorded among migrants. Of these, 5 patients (17.2%) had isoniazid resistance, and 1 strain (3.4%) was classified as multidrug-resistant (MDR). The patients with isoniazid-resistant TB were originally from Romania (2), Venezuela (1), Bolivia (1), and India (1) the MDR patient was from Romania. Cases were detected sporadically during the study period with no apparent time clustering.

Regarding treatment outcomes, no statistically significant differences were observed between groups in terms of cure rates, mortality, or hospital admission. Two patients (5.7%) in the native-born group and four (6.9%) in the migrant group died (*p* = 1). Hospitalization was required in 33 cases (47.8%) among non-migrants and 13 cases (44.8%) among migrants (*p* = 1).

Loss to follow-up was more frequent in the migrant group than in the native group: 3 patients (10.3%) vs. 2 patients (2.9%), respectively (*p* = 0.152). In both groups, 10% of patients experienced treatment failure: 7 cases among native-born individuals and 3 among migrants.

### 3.3. Analysis of Unfavourable Outcomes and Mortality

When analyzing the unfavourable outcome variable (Table 2), we found statistically significant differences by sex: all 11 cases (100%) of unfavourable outcomes occurred in male patients (*p* = 0.014).

A significant association was found between elevated C-reactive protein (CRP) levels and unfavourable outcomes (*p* = 0.033). However, no statistically significant association was found with serum albumin levels (*p* = 0.065), comorbidities, or substance use. The CRP/albumin ratio was significantly associated with unfavourable outcomes (*p* = 0.015) (Figure 2).

A ROC curve was generated to evaluate the optimal discriminatory threshold. The best cutoff value was determined to be 12.44, which yielded a sensitivity of 72%, specificity of 60%, and an area under the curve (AUC) of 0.709. In the multivariable analysis, the only parameter that retained significance was the CRP/albumin ratio, showing an odds ratio (OR) of 4.08 (95% CI 1.09–19.62; *p* = 0.048).

We also found a significant association between mortality and the CRP/albumin ratio (*p* = 0.014) (Figure 3). In this case, the optimal AUC threshold was 29.42, with an odds ratio of 3.15 in the logistic regression model (95% CI 0.83–11.65; *p* = 0.08).

## 4. Discussion

In this study, migrant patients accounted for 29% of all TB cases diagnosed in the Elda Health Department. Compared with native-born patients, they were generally younger, reported lower rates of alcohol, tobacco, and other substance use, and had fewer comorbidities. However, they exhibited a higher prevalence of resistance to first-line anti-tuberculosis drugs and a trend toward higher loss-to-follow-up rates.

The proportion of migrant TB patients observed here is lower than the 46.8% reported by the Carlos III Health Institute in 2023 [14], likely reflecting the smaller proportion of migrants in the Elda Health District compared with other regions included in national studies.

Sex distribution in our cohort was consistent with previous reports, with TB more prevalent in men than in women in both migrant and native-born populations [1,5,8,14,15]. The significantly younger age profile of foreign-born patients compared with native-born patients [14,15] aligns with demographic patterns, as most migration typically occurs during younger adulthood [15].

The primary site of disease was pulmonary TB, consistent with findings from previous studies [5,8,14,15]. Native-born patients presented with a higher number of comorbidities, likely related to their older age, although the difference was not statistically significant. According to the National Epidemiological Surveillance Network, 7.4% of TB cases in Spain are co-infected with HIV—a proportion similar to that observed in our native-born patient group [8,14]. This incidence has declined in recent years, likely reflecting improved access to antiretroviral therapy [1].

Regarding treatment outcomes, no significant differences were found between migrant and native-born groups in cure rates, hospital admissions, or mortality. The overall mortality rate was 6%, in line with previous reports [14]. Although not statistically significant, loss to follow-up occurred more frequently among migrants, mainly due to return to their country of origin. This is a public health concern given the associated risk of ongoing transmission and the potential emergence of new drug-resistant strains [6,16].

Analysis of the composite variable “unfavourable outcome” revealed a significant association with male sex, as all cases of mortality or loss to follow-up occurred in men [14]. Unfavourable outcomes were also significantly associated with elevated CRP levels, and both unfavourable outcomes and mortality were linked to a higher CRP/albumin ratio.

Previous studies have shown that hypoalbuminemia, low serum protein levels, and malnutrition are associated with poor prognosis in TB patients, particularly in intensive care settings [17,18]. The CRP/albumin ratio is of particular interest because it reflects both the inflammatory burden of disease and the patient’s nutritional and functional status. This dual representation may enhance its prognostic value compared with single-parameter markers. For example, while other inflammatory indices such as the neutrophil-to-lymphocyte ratio have been linked to adverse outcomes in infectious diseases, the CRP/albumin ratio may capture a broader pathophysiological profile by integrating markers of inflammation with baseline physiological reserve. These features suggest that it could serve as a practical, low-cost, and widely accessible tool for mortality risk stratification and to support clinical decision-making in TB management.

Alcohol and tobacco use are major global risk factors for TB, and patients engaging in these behaviours have higher rates of treatment failure [16]. This association likely reflects both the immunosuppressive effects of substance use and the social factors that often accompany it. Malnutrition and social marginalization further increase susceptibility to TB infection [16].

In our cohort, substance use was more common among native-born patients than among migrants, a pattern that may be partly explained by the “healthy migrant effect” [16,19]. Clear sex-based differences have also been reported in substance use, with men more frequently engaging in these behaviours [13].

We identified nine cases of drug-resistant TB, six of which (20.8%) occurred in the migrant population. Isoniazid resistance was notably high, affecting five migrant patients (17.2%), and one case (4.34%) met the criteria for multidrug-resistant TB (MDR-TB). Recent data indicate an increase in drug resistance in low-burden countries, particularly in areas with rising migration from high-burden TB regions [20].

The MDR-TB case in our study involved a migrant from Romania, a country with a high MDR-TB prevalence. This patient was lost to follow-up after returning to his country of origin, and his outcome remains unknown. This underscores the need for heightened awareness of the potential for increased drug resistance following major migration events from high-incidence settings. Ensuring high-quality care and preventing the spread of MDR-TB strains must be priorities [5].

Transnational care strategies—such as standardized treatment records, cross-border data sharing, and coordination with health authorities in patients’ countries of origin—could help ensure continuity of care. While relevant worldwide, these measures may be particularly critical within the EU, where patient mobility across member states is common.

Finally, universal access to molecular diagnostic tools is essential, particularly for the rapid detection of resistance mutations, at a minimum those conferring rifampicin resistance.

Fifty-five percent of migrants in our cohort had been living in Spain for less than five years. This aligns with reports that TB diagnosis in migrant populations often occurs within the first two years after arrival in the host country [5,6], a pattern closely linked to poverty, overcrowded housing, and other adverse living conditions. Greater socioeconomic vulnerability among migrants may also facilitate TB reactivation or reinfection [21]. Some studies indicate that the risk of TB remains elevated for more than 10 years after migration [5,21].

Current screening policies in Spain follow European Centre for Disease Prevention and Control recommendations, focusing on targeted testing of high-risk groups, with particular emphasis on recent arrivals [22]. However, a growing body of evidence shows that TB cases also occur well beyond the initial years post-arrival, suggesting that screening and prevention strategies should be extended beyond the current timeframe.

Strengths of this study include its population-based design, encompassing all known TB cases among both migrant and native-born individuals in the Elda Health Department over the past decade. Limitations include the retrospective design, which carries a risk of missing or incomplete data, and the fact that the study population may not be fully representative of migrant populations in other geographic settings. Nonetheless, a comprehensive search across multiple data sources was performed to minimize these limitations.

## 5. Conclusions

Tuberculosis among migrants accounted for 29% of all cases diagnosed in the Elda Health Department. In both groups, prevalence was higher among men, but migrant patients were younger, reported lower rates of substance use, tended to have fewer comorbidities, and experienced higher rates of loss to follow-up compared with native-born patients. Migrants also showed a higher frequency of drug resistance, supporting the recommendation to perform comprehensive drug susceptibility testing on *Mycobacterium tuberculosis* isolates from this population.

Romania was the most common country of origin, and more than half of migrant patients had arrived in Spain within the previous five years. No significant differences were observed between migrants and native-born individuals in cure rates, hospital admissions, or mortality. Unfavourable outcomes were associated with a higher CRP/albumin ratio.

## Figures and Tables

**Figure 1 pathogens-14-00824-f001:**
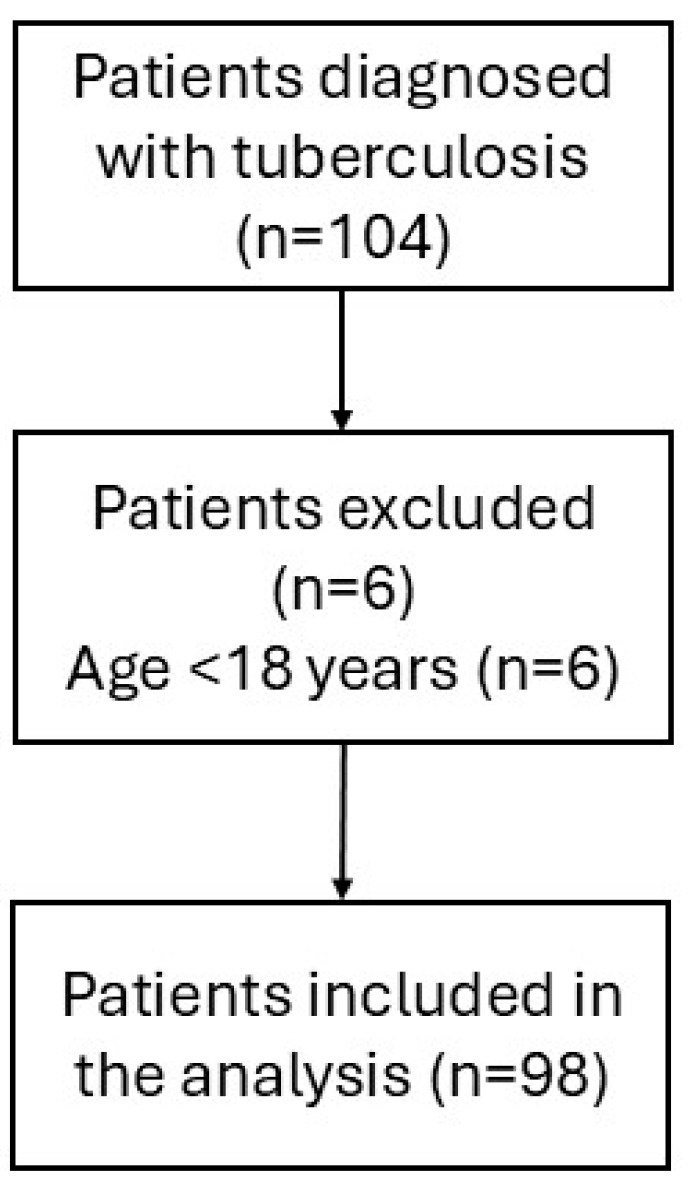
Flow diagram of patient inclusion and reasons for exclusion.

**Figure 2 pathogens-14-00824-f002:**
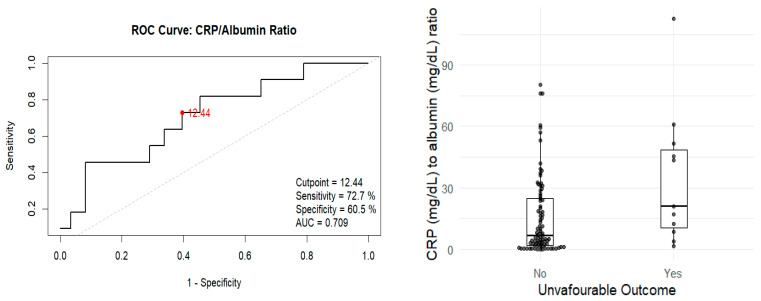
Association between unfavourable outcome and CRP/albumin ratio. Receiver operating characteristic (ROC) curve of the CRP/albumin ratio for predicting unfavourable outcomes (mortality, treatment failure, or loss to follow-up) in patients with tuberculosis. The optimal cut-off point (12.44) is indicated, corresponding to a sensitivity of 72% and specificity of 60% (AUC = 0.709).

**Figure 3 pathogens-14-00824-f003:**
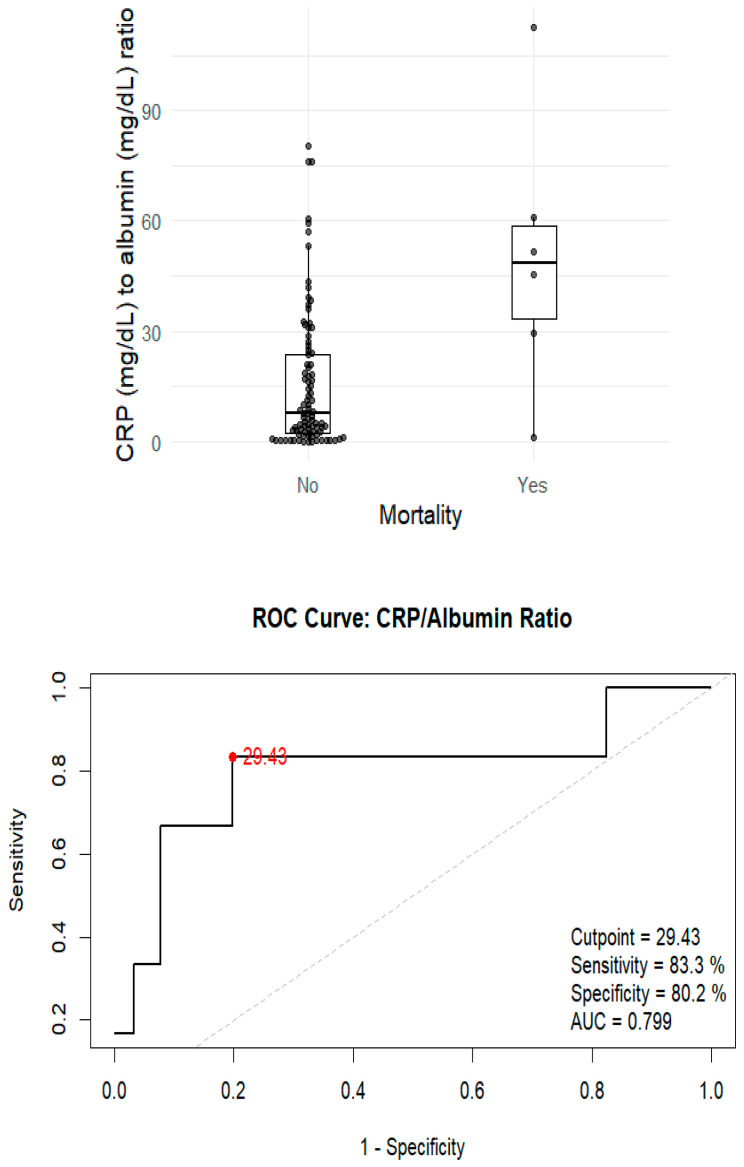
Association between mortality and CRP/albumin ratio. ROC curve of the CRP/albumin ratio for predicting in-hospital mortality in patients with tuberculosis. The optimal cut-off point (29.42) is indicated, with its corresponding sensitivity and specificity.

**Table 1 pathogens-14-00824-t001:** Clinical and epidemiological characteristics of tuberculosis in migrants and native patients.

Variables	Total (n = 98)	Native Population (n = 69)	Migrant Population (n = 29)	*p*-Value
Time in Spain < 5 years	16 (16%)	0	16 (55.2%)	0.001
Sex				
Male	66 (67%)	47 (68%)	19 (65%)	0.988
Female	32 (32.6%)	22 (31%)	10 (34.5%)	
Mean age (SD) (years)	44.38 (17.6)	48.03 (17.6)	35.69 (14.3)	0.001
Pulmonary TB	81 (82%)	57 (82%)	24 (82%)	1
Extrapulmonary TB	17 (17%)	12 (17%)	5 (17%)	
COPD	10 (10%)	9 (13%)	1 (3.45%)	0.273
Heart failure	1 (1%)	1 (1.4%)	0	1
AMI/Stroke	3 (3.5%)	3 (4.35%)	0	0.553
Cancer/Autoimmune	14 (14.2%)	12 (17.3%)	2 (6.9%)	0.22
Neurologic disease	9 (9.18%)	6 (8.7%)	3 (10.3%)	1
HIV infection	6 (6.1%)	5 (7.25%)	1 (3.45%)	0.667
Diabetes Mellitus	7 (7.1%)	5 (7.25%)	2 (6.9%)	1
Toxic habits				
Tobacco use	56 (57%)	46 (66%)	10 (34%)	0.007
Alcohol consumption	35 (35%)	32 (46%)	3 (10.3%)	0.001
IV drug use	7 (7.1%)	7 (10%)	0	0.1
Albumin (g/dL, SD)	3.91 (0.66)	3.91 (0.66)	3.90 (0.68)	0.997
CRP (mg/L, SD)	59.71 (64.5)	56.73 (60.6)	66.8 (73.4)	0.929
Educational level				
Higher education (University degree or equivalent post-secondary qualification)	5 (5%)	5 (7.24%)	0	0.33
Secondary education (Completion of high school or equivalent vocational training)	50 (50%)	35 (50%)	15 (51%)	
Primary education (Completion of primary/elementary school only)	35 (35%)	25 (36%)	10 (34%)	
Unknown (Educational level not reported or unavailable)	8 (8%)	4 (5.7%)	4 (13%)	
Employment status				
Employed	57 (58%)	42 (60.8%)	15 (51.7%)	0.595
Unemployed	31 (31%)	19 (27.5%)	12 (41%)	

SD: Standard deviation, TB: Tuberculosis, COPD: Chronic obstructive pulmonary disease, AMI: Acute myocardial infarction.

**Table 2 pathogens-14-00824-t002:** Association between unfavourable outcome and clinical-epidemiological variables.

Variable	Unfavourable Outcome: Yes	Unfavourable Outcome: No	*p*-Value
Sex			
Male	11 (100%)	55 (63%)	**0.01**
Female	0	—	
Mean age (SD), years	46.3 (20.9)	44.1 (17.3)	0.75
Origin			
Migrant	4 (14%)	25 (86%)	0.72
Native	7 (10%)	62 (90%)	
Comorbidities			
COPD	2 (18%)	9 (82%)	0.31
Heart failure	0	11 (100%)	1
AMI/Stroke	0	11 (100%)	1
Cancer/Autoimmune disease	0	11 (100%)	0.35
Neurologic disease	2 (18%)	9 (82%)	0.26
HIV infection	1 (9%)	10 (91%)	0.52
Toxic habits			
Tobacco use	6 (55%)	5 (45%)	0.89
Alcohol use	4 (36%)	7 (64%)	1
IV drug use	7 (10%)	0	0.1
Drug resistance			
Isoniazid resistance	0	11 (100%)	1
Laboratory values			
Albumin (g/dL), median (IQR)	3.6 (2.78–4.15)	4.11 (3.5–4.4)	0.065
CRP (mg/L), median (IQR)	102 (38.9–150)	25 (9.95–85.5)	0.033
BMI (kg/m^2^), median (IQR)	18.4 (18.2–18.7)	22.6 (18.7–26.9)	0.127
CRP/Albumin ratio	20.8 (10.5–48.6)	6.72 (2.13–24.7)	0.024

SD: Standard deviation, COPD: Chronic obstructive pulmonary disease, AMI: Acute myocardial infarction. IQR: Interquartilic range, CRP: C-reactive protein, BMI: body mass index.

## Data Availability

The original data presented in the study are openly available in https://github.com/pwjpwj/Imported-authoctonous-TBC, accessed on 5 August 2025.

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
