# Peer review of "Human Tuberculosis in Migrant and Autocthonous Patients: A Ten-Year Single-Centre Experience"

_pathogens, 2025, doi:10.3390/pathogens14080824_

Round 1
Reviewer 1 Report
Comments and Suggestions for Authors
Dear Editor,
Thank you for the opportunity to review “Imported and Autochthonous Tuberculosis a Ten-Year Single Centre Experience”. The manuscript offers an interesting regional perspective of human tuberculosis cases in Spain from the view of the migrant and non-migrant status of the patients. The scientific analysis is adequate; however, I suggest the following changes to improve the readability of the manuscript.
Major:
The results are interesting but as presented in the text are really confusing; I suggest rewriting the result section considering Table 1 as a reference and disclosing for each variable the result for the migrant and non-migrant population, including statistical significances if any. As presented in this section, data is not organized and very difficult to follow – I suggest increasing readability in this section.
Title: This journal publish human and animal cases, please rephrase to: “Imported and Autochthonous Human Tuberculosis Cases in Elda Health District, Alicante, Spain, 2013-2023”
Minor:
L15: Please rephrase this sentence – reads awkward
L17: Please include region if applicable, and country. Delete ‘Methods’
L15-19: several concepts are repeated, please summarize
L19: Delete “results”
L20-21: Why is the epidemiologic data of the imported cases relevant but not the autochthonous?
L25: p should be italicized
L26: Delete ‘conclusion’
L26: “Tuberculosis in the migrant population accounted for 29% of total cases” is a result, not a conclusion
L36: That is not accurate as there are zoonotic cases by M. bovis and M. caprae in Spain, -and by ‘atypical’ mycobacteria. I suggest disclosing this information briefly
L75: I suggest reorganizing this section into: Ethical Considerations, Inclusion Criteria, Diagnostic Workup, Statistical Analysis
L84-85: Repetitive
L92-95: Include results for this, including the species of Mycobacterium in the result sections
L102: Define WHO
L130: Is this for both groups or autochthonous only?
Reviewer 2 Report
Comments and Suggestions for Authors
Dear Authors,
Thank you for submitting your manuscript to Pathogens. Your study addresses a timely and important issue: tuberculosis among migrant and native-born populations, and provides valuable local data from the Elda Health Department. The retrospective cohort design is appropriate, and the focus on drug resistance and CRP/albumin as prognostic indicators is particularly relevant.
However, I have some comments and suggestions aimed at strengthening the clarity, methodological rigor, and clinical impact of your work:
Study Design and Definitions:
Please clarify the operational definitions of “imported” vs. “autochthonous” tuberculosis. Is the categorization based on place of birth, nationality, or time since migration?
Include the percentage of migrants in the Elda Health District during the study period to contextualize the 29% TB burden.
Clarify whether ethics committee approval was obtained or waived, and name the relevant committee.
Methodology and Statistical Analysis:
Explain how missing data were handled. Were any imputation methods used?
Indicate how patients with incomplete records were managed in statistical analyses.
Provide rationale for the choice of cutoff values in the CRP/albumin ratio (e.g., use of Youden index?).
Report confidence intervals consistently for key percentages and odds ratios.
Consider adding a survival analysis (e.g., Kaplan–Meier) if time-to-event data are available.
Results Presentation:
Break the Results section into subsections for clarity (e.g., Demographics, Drug Resistance, Outcomes).
Highlight trends that are clinically relevant even if not statistically significant, such as loss to follow-up in migrants.
Clarify whether drug-resistant TB cases clustered by country or time.
If data permit, describe any common features among patients lost to follow-up.
Drug Resistance and Laboratory Testing:
Clarify whether resistance testing included second-line drugs and was performed routinely or selectively.
Indicate if any molecular epidemiological data (e.g., spoligotyping, MIRU-VNTR) were available, or acknowledge if outside the scope of this study.
CRP/Albumin Ratio and Prognostic Value:
The association of this ratio with mortality and treatment outcomes is compelling. Please discuss its clinical utility as a prognostic marker and how it compares with other markers (e.g., neutrophil-to-lymphocyte ratio).
Provide ROC curves with thresholds annotated and improve figure legends accordingly.
Public Health and Policy Implications:
Discuss whether the current TB screening policies in Spain (at time of data collection) adequately address the risks found in this study.
Offer specific recommendations for mitigating higher loss to follow-up in migrants, including transnational care strategies.
Comment on implications for systematic screening beyond 2–5 years post-arrival, especially in light of high resistance prevalence.
Finally, I also have some minor comments for your consideration:
Use “migrant” or “foreign-born” terminology instead of “imported TB” throughout the manuscript for consistency with international standards.
Define all abbreviations in tables (e.g., AMI, IDU, COPD).
Highlight statistically significant results in tables using bold or asterisks for visibility.
Improve the abstract by simplifying phrasing and clarifying the meaning of “unfavourable outcome.”
Reword long or repetitive sentences throughout for improved flow (e.g., Results and Discussion).
Include a flow diagram (e.g., STROBE) to show patient inclusion/exclusion.
Ensure consistency in percentage formatting (one or two decimals throughout).
Specify whether time since migration was recorded as a continuous or categorical variable.
Clarify employment and education status categories if available.
Improve readability of figures by increasing label size and adding interpretation cues.
Comments on the Quality of English Language
The manuscript is generally understandable and written in competent academic English. However, there are areas where the language could be improved to enhance clarity and flow. Some sentences are overly long or repetitive, and a few grammatical or stylistic inconsistencies are present.
Round 2
Reviewer 2 Report
Comments and Suggestions for Authors
Dear authors:
The revised manuscript demonstrates substantial improvement in clarity, methodological transparency, and alignment between results and conclusions. The introduction now provides a broader and more precise context, with relevant epidemiological background and up-to-date references, including international guidance. The definition of “migrant” and inclusion/exclusion criteria are clearly stated, and the description of the study setting now includes the proportion of migrants in the population.
The methods section is thorough, with clear explanations of laboratory procedures, drug susceptibility testing (including second-line DST criteria), handling of missing data, and statistical approaches such as the ROC cut-point determination. Ethics approval and informed consent considerations are now fully documented.
The results are better organized into logical subsections, supported by a flow diagram, detailed tables with defined abbreviations, and improved figure legends. Non-significant but clinically relevant trends are highlighted appropriately. The addition of country-specific drug resistance data and notes on temporal clustering strengthens the interpretation.
The discussion has been expanded to address the prognostic value of the CRP/albumin ratio, compare it with other inflammatory markers, and consider public health implications, including recommendations for extended TB screening periods and transnational care strategies for mobile populations. The conclusions are now well-supported by the data and clearly linked to practical recommendations.
The quality of English is clear and professional, with improved flow and standardized terminology. Figures and tables are well-presented, and all abbreviations are defined.